# Synchrotron Radiation Refraction-Contrast Computed Tomography Based on X-ray Dark-Field Imaging Optics of Pulmonary Malignancy: Comparison with Pathologic Examination

**DOI:** 10.3390/cancers16040806

**Published:** 2024-02-16

**Authors:** Eunjue Yi, Naoki Sunaguchi, Jeong Hyeon Lee, Seung-Jun Seo, Sungho Lee, Daisuke Shimao, Masami Ando

**Affiliations:** 1Department of Thoracic and Cardiovascular Surgery, Korea University Anam Hospital, Seoul 02841, Republic of Korea; sholeemd@korea.ac.kr; 2Department of Radiological and Medical Laboratory Sciences, Graduate School of Medicine, Nagoya University, Nagoya 461-8673, Japan; sunaguchi@met.nagoya-u.ac.jp; 3Department of Pathology, Korea University Anam Hospital, Seoul 02841, Republic of Korea; pathjhlee@gmail.com; 4Department of Experimental Animal Facility, Daegu Catholic University Medical Center, Daegu 42472, Republic of Korea; bmejun@gmail.com; 5Faculty of Health Sciences, Butsuryo College of Osaka, Osaka 593-8328, Japan; daisuke.shimao@gmail.com; 6Photon Factory, Institute of Materials Structure Science, High-Energy Accelerator Research Organization, Tsukuba 300-3256, Japan; msm-ando@rs.noda.tus.ac.jp

**Keywords:** refraction contrast, lung cancer, diagnosis

## Abstract

**Simple Summary:**

The imaging method called refraction-contrast computed tomography using synchrotron radiation can achieve detailed images almost similar to those from a microscope. This study examined its ability to recognize lung cancer. We took pictures of lung cancer samples in Japan using this imaging method. These pictures showed the insides of the samples precisely, making it easy to distinguish healthy tissue from cancerous spots. They even allowed us to tell the difference between a main lung cancer and one that started somewhere else, like the colon. Fundamentally, this method could be a way to check for lung cancer without cutting into tissue. However, before doctors can use it, the machines involved need some major changes.

**Abstract:**

Refraction-contrast computed tomography based on X-ray dark-field imaging (XDFI) using synchrotron radiation (SR) has shown superior resolution compared to conventional absorption-based methods and is often comparable to pathologic examination under light microscopy. This study aimed to investigate the potential of the XDFI technique for clinical application in lung cancer diagnosis. Two types of lung specimens, primary and secondary malignancies, were investigated using an XDFI optic system at beamline BL14B of the High-Energy Accelerator Research Organization Photon Factory, Tsukuba, Japan. Three-dimensional reconstruction and segmentation were performed on each specimen. Refraction-contrast computed tomographic images were compared with those obtained from pathological examinations. Pulmonary microstructures including arterioles, venules, bronchioles, alveolar sacs, and interalveolar septa were identified in SR images. Malignant lesions could be distinguished from the borders of normal structures. The lepidic pattern was defined as the invasive component of the same primary lung adenocarcinoma. The SR images of secondary lung adenocarcinomas of colorectal origin were distinct from those of primary lung adenocarcinomas. Refraction-contrast images based on XDFI optics of lung tissues correlated well with those of pathological examinations under light microscopy. This imaging method may have the potential for use in lung cancer diagnosis without tissue damage. Considerable equipment modifications are crucial before implementing them from the lab to the hospital in the near future.

## 1. Introduction

The pulmonary parenchyma has a unique composition and includes pulmonary arterioles, venules, bronchioles, alveolar sacs, and interalveolar septa [1]. In living organisms, most pulmonary spaces are filled with air, which makes the viscera and vasculature appear light compared to the dark background in traditional X-ray roentgenography [2].

At the onset of certain pulmonary diseases, the interstitium and alveolar sac suffer from distortion as the cells in the alveolar septum or interstitial spaces proliferate inappropriately or produce secretions, often leading to consolidative lesions. Early malignancies arising from fine alveolar structures appear on imaging as ground-glass opacity (GGO) [3]. GGO resembles a small hazy fog that does not involve other microstructures, including bronchioles and terminal venules.

These minute infiltrative changes in the interstitial area of normal lung tissue can be benign [4], inflammatory, and temporary; however, such lesions are often considered to represent early-stage lung cancer, especially when they persist for >3 months [5]. Conventional chest tomography can detect and distinguish an abnormal from a normal parenchyma; however, pathological examination with sequential histology is required for diagnostic confirmation [6].

Although the histopathological examination of lung tissue under light microscopy has been the traditional method for definitive diagnosis, resected specimens must be divided into serial sections after fixation with formalin or alcohol because of the limited penetration of optical microscopy, creating irreversible damage to the tissue sample. The volume shrinkage after fixation is estimated to be 3–6 vol% in formalin and up to 20 vol% in alcohol. Only two-dimensional pathological images can be observed after histological processing [7], and the actual dimensions and shapes of the lesions are unknown. 

Remarkable advances have been made in the structural analysis of alveolar sacs by using synchrotron radiation tomography of human lung tissue [8,9,10]. Precise analysis was performed, including three-dimensional (3D) analysis of a normal alveolar sac. The specimens used for the experiments were prepared very carefully, maintaining the in vivo morphology during harvest and successfully reconstructing the 3D microstructure; however, the use of a fixative and a slight volume change were unavoidable. 

The method of refraction-contrast computed tomography based on X-ray dark-field imaging (XDFI) using synchrotron radiation can be beneficial for understanding tissue organization, with 3D reconstruction images supported by higher contrast and spatial resolution than conventional X-ray imaging, comparable to pathological examination [11,12,13]. The monochromatic and high-beam parallel characteristics of synchrotron X-rays produce images that can distinguish soft tissues with similar attenuation coefficients. This imaging method is efficient in soft-tissue discrimination, including articular cartilage, human breast, eyeballs, and coronary arteries [14,15,16,17].

We have accumulated tomographic images of various pulmonary lesions using this method since 2019 in the Photon Factory BL14B of the High-Energy Accelerator Research Organization in Tsukuba, Japan [18]. In this study, synchrotron radiation tomographic images of two types of human lung cancer (two primary and two secondary adenocarcinoma samples) were acquired, reconstructed in 3D, and compared with images from pathological examinations, thereby identifying potential implications for clinical applications of the XDFI method.

The work of our experiments can be described as follows. (1) The acquisition of SR images using XDFI optics from human lung adenocarcinomas: either primary or secondary. (2) The specifications of secondary adenocarcinoma: the acquisition of SR tomographic images of secondary adenocarcinoma, originating from rectal cancer. (3) Comparison with pathology: We conducted a comparison of these images with pathological examinations. (4) The exploration of potential: We explored the potential of refraction-contrast SR CT images in discriminating between different types of pulmonary malignancies.

Recently, we proposed new imaging methods for improving spatial resolution by placing a scintillator in close contact with a Laue angle analyzer (LAA), thus eliminating the distance required for X-ray interference and wavefront separation. Further experiments with lung cancer specimens will be conducted using this method [19].

## 2. Materials and Methods

### 2.1. Tissue Preparation

Lung tissues for imaging acquisition were donated by patients who had undergone surgery for the treatment of malignancy at Korea University Anam Hospital; written informed consent was obtained from all patients. Specimens were harvested under the supervision of a pathologist specializing in pulmonology (J.H. Lee) after all necessary diagnostic procedures were completed. The lung specimens used in this study are listed in Table 1. This study was approved by the Institutional Review Board of the Korea University Anam Hospital (IRB number: 2019AN0242).

The usual pathological procedure included airway inflation, fixation with 10% neutral-buffered formalin solution, gross examination, the mapping of cancer lesions, serial sectioning in 3 mm increments, paraffin block fixation, hematoxylin and eosin staining, and immunohistochemistry. Specimens for SR imaging were harvested, placed in a 1% agarose gel solution for transportation, and brought to the BL14B Photon Factory of the High-Energy Accelerator Research Organization (Tsukuba, Japan).

### 2.2. X-ray Source and Experimental Setup

The refraction-contrast XDFI-based computed tomography (CT) imaging system was constructed on beamline BL14B at the High-Energy Accelerator Research Organization Photon Factory in Tsukuba, Ibaraki, Japan. In this system, synchrotron radiation generated from a 5-Tesla superconducting vertical wiggler of BL14 was used as the incident X-ray source. The X-ray beam was monochromatized using a double-crystal monochromator located outside the beamline and injected into the imaging system inside the beamline.

The imaging system consisted of an asymmetrically cut monochromator collimator (AMC) made from an asymmetric Si single crystal, which is denoted as (A) in Figure 1, an acrylic cylindrical filter (B), a sample rotation stage (C), a Laue-type angle analyzer (LAA) made from thin Si single-crystal plates (D), an X-ray scintillator (E), lens optics (F), and an X-ray camera (G). Because the X-ray beam was generated by a vertical wiggler, the polarization direction of the beam was perpendicular to the ground, allowing the crystal optics in the imaging system to be placed horizontal to the ground. The experimental setup is illustrated in Figure 1.

The use of an AMC led to an increase in the beam width of the X-ray beam and a reduction in the divergence angle of the diffracted beam compared with that of the incident beam. These effects can be controlled by the b factor (b = sin(θ_B_ − α)/sin(θ_B_ + α)) of the AMC, where θ_B_ and α are the Bragg angle and the asymmetry angle, respectively. The X-ray beam propagating through the sample was split into two directions: forward diffraction and diffraction at the LAA. By placing an X-ray camera in the forward diffraction direction, a refraction-contrast image that expressed the refraction angle of the X-ray beam generated in the sample could be obtained. The acrylic cylindrical filter was an acrylic block with a hole of the same diameter as that of the specimen container, which can flatten the absorption contrast produced by the specimen container, thereby significantly compensating for the large refractive effect at the cylinder limb. The specimen was fixed in a cylindrical container using agarose gel with an absorption coefficient close to that of the specimen, and the absorption contrast produced by the specimen was negligible.

### 2.3. Acquisition and Reconstruction of Images

The specimen was placed on a sample stage in an experimental hutch controlled by a step motor. Lung specimens were placed in a 2 cm diameter cylinder with 1% agarose gel to prevent movement during image acquisition (Figure 2). Table 2 lists the imaging conditions used in this study. Among the typical parameters, the X-ray energy was 19.8 keV, the fields of view of the X-rays after diffraction by the AMC were approximately 23 mm (horizontal) and 21 mm (vertical), and the horizontal divergence of the beam was approximately 0.5 arcsec. The thickness of the LAA was 166μm. The LAA was propped against a mirror-polished Si plate to prevent distortion. The pixel size of the X-ray camera was 5.5 μm. The spatial resolution of the system was approximately 10 μm.

Our imaging method can provide high-resolution volumetric images of lung specimens. We obtained 3D tissue images using the 3D image processing software Amira-Avizo (version 2020.3), developed by Thermo Fisher Scientific Inc. (Visualization Sciences Group, Burlington, MA, USA), and 3D-rendered tomographic images were produced using a graphic processing unit.

### 2.4. Comparison with Pathologic Examinations

After image acquisition, the specimens were transported to the hospital and embedded in paraffin blocks for pathological examination. Pathological slides from paraffin blocks were stained with hematoxylin and eosin and observed under a light microscope by a specialized pathologist. Images obtained under a light microscope were compared with those obtained using refraction-contrast CT. Table 3 compares the imaging conditions among examinations conducted using light microscopy, μ-CT, and synchrotron radiation based on XDFI optics.

## 3. Results

### 3.1. Images from Primary Lung Adenocarcinoma

Two pulmonary adenocarcinoma specimens were used for the imaging. One was primary lung cancer without a lepidic pattern (specimen #1, Figure 3), and the other had a lepidic pattern (specimen #2, Figure 4).

### 3.2. Lung Adenocarcinoma without Lepidic Pattern

In the images of specimen #1, a similar consolidative lesion can be observed in both the pathological examination under light microscopy (×10 magnification, Figure 3a) and SR CT (Figure 3b). The spiculate margin between the malignant lesion and the normal area is clearly demarcated. The 3D volume rendering and segmentation images (Figure 3c,d) enabled the structural observation of the cancer (Appendix A).

**Figure 3 cancers-16-00806-f003:**
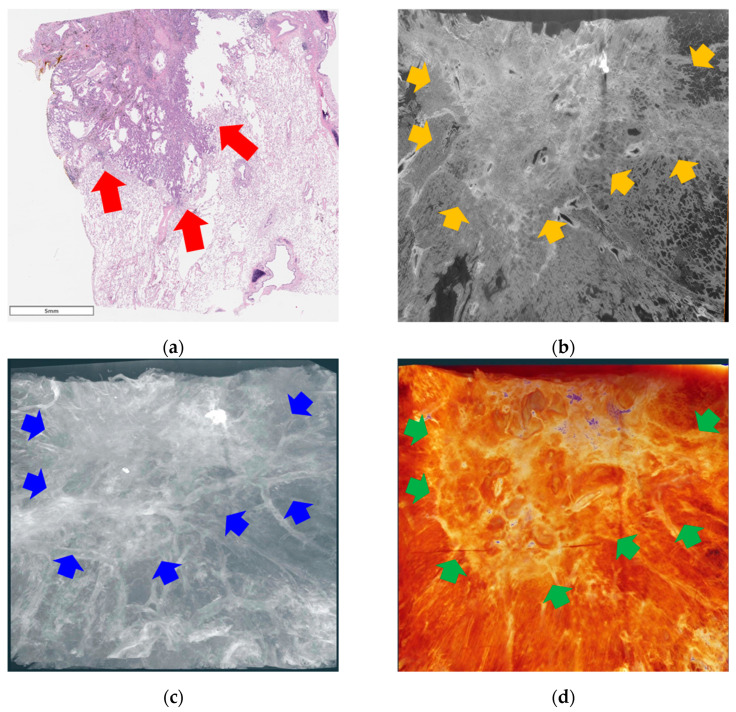
Images of specimen #1. (**a**) Pathological examination under a light microscope (scanning view). A malignant lesion consisting of a solid component (red arrows). (**b**) One-slice image of the reconstructed tomographic volume. The area of cancer (orange arrows) is similar to that of the pathological specimen. Eccentric calcifications and bubble-like lucencies in the lesions. (**c**) 3D image from reconstructed tomographic volume. Cancer lesions containing eccentric calcifications and bubble-like lucencies (blue arrows). (**d**) 3D image produced by volume rendering of the reconstructed tomographic volume. The border of the tumor with spiculated spikes (green arrows) is well demarcated from the normal area. The three-dimensional aspects of the tumor lesions observed through whole video images (Appendix A).

### 3.3. Lung Adenocarcinoma with a Lepidic Pattern

Part of specimen #2 contained a lepidic pattern, which is a specific pathological finding of early-stage adenocarcinoma. In this area, malignant changes in the nucleus (irregular shape, multiple nucleoli, and enlarged nucleoli) were observed; however, the framework of the alveolar structure could still be identified. Pathological examination revealed a thickened alveolar wall (Figure 4a, inner circle with blue dots); a similar shape was observed in the refraction-contrast CT image (Figure 4b). In the 3D volume rendering and segmentation images, the area of the lepidic pattern is identifiable with fog-like haziness containing the alveolar microstructure (Figure 4c,d). An irregular spiculate margin around the cancerous area can also be observed, similar to that observed in specimen #1. The 3D structure of this tumor is well expressed in the volume-rendered images (Appendix A).

**Figure 4 cancers-16-00806-f004:**
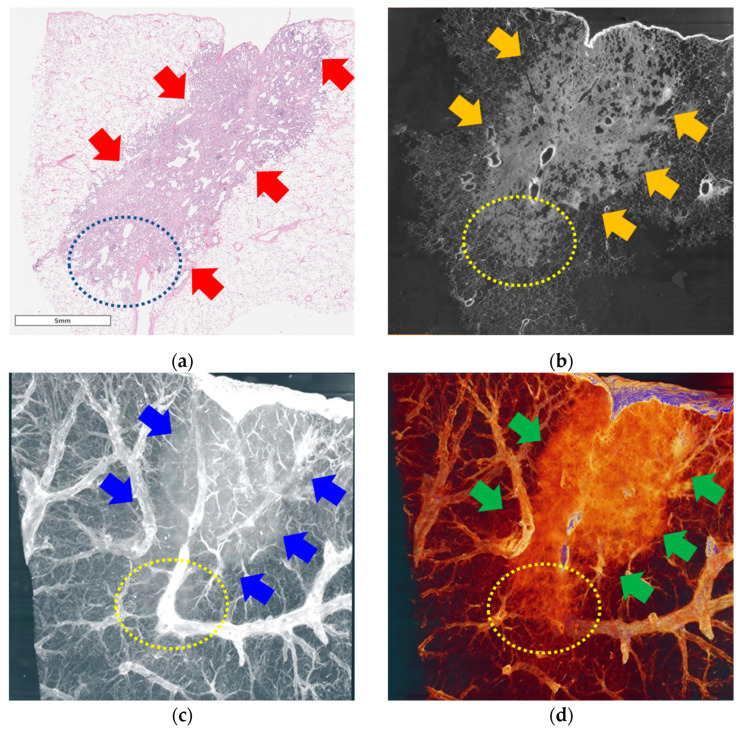
Images of specimen #2. (**a**) Pathological examination under a light microscope (scanning view). A malignant lesion consisting of a solid component (red arrows). Specimen exhibiting a lepidic pattern (blue dotted circle), which was thought to be the source of the ground-glass opacity (GGO) on tomography. (**b**) One-slice image of the reconstructed tomographic volume. The area of cancer (orange arrows) is similar to that observed on pathological examination. Eccentric calcifications and bubble-like lucencies are observed in the lesions. Ground-glass opacity in the lower parts of the cancerous lesion (yellow dotted circle). (**c**) 3D image from reconstructed tomographic volume. A cancerous lesion (blue arrow) containing the GGO portion (yellow arrow). (**d**) 3D image produced by volume rendering of the reconstructed tomographic volume. The border of the tumor with spiculated spikes (green arrows) and the ground-glass opacity (GGO) portion are well demarcated from the normal lung area. The three-dimensional aspects of the tumor lesions can be observed in whole video images (Appendix A).

### 3.4. Images from Secondary Lung Adenocarcinoma

Two specimens were harvested from patients with secondary lung adenocarcinoma (of colorectal origin). The margin between the malignant lesion and the normal area was round compared to that of the primary lesion. Irregular spikes were rarely observed in the images of the two lung tissues (sample #3; Figure 5 and sample #4; Figure 6). Secretions from cancer cells (the pinkish lesion in pathological examination, Figure 5a) appeared as multiple whitish small dots inside the tumor in the refraction-contrast CT image (Figure 5b), which was also observed in both 3D rendering and segmentation images (Figure 5c,d).

In the image of specimen #4, multiple pore areas containing small malignant lesions can be observed (red arrows in Figure 6a). Similar findings can be observed in the 3D images (Figure 6c,d, blue and green arrows); however, the malignant area was not clearly demarcated because the section was thought to be inside the pore that melted away during the preparation process. The lepidic pattern observed in one of the primary lung adenocarcinomas was not observed in any of the secondary lung adenocarcinomas.

### 3.5. Three-Dimensional Volume Estimation

We performed 3D volume reconstruction using Amira-Avizo software (version 2020.3, Visualization Sciences Group, Burlington, MA, USA). Figure 3c, Figure 4c, Figure 5c and Figure 6c show one cross-section in the 3D volume-rendering images, and Figure 3d, Figure 4d, Figure 5d and Figure 6d show segmentation using this software. Two oblique views of the 3D images and were compared for each specimen to estimate the tumor volume (Figure 7). The exact depth and area occupied by the specimens were identified, which was not possible through pathological examination under a light microscope.

## 4. Discussion

In our study, we acquired refraction-contrast SR CT images of two different types of lung adenocarcinoma: primary and secondary (metastatic cancer originating from rectal cancer). Through comparison with pathological examinations, we explored the potential of refraction-contrast SR CT images for discriminating different types of pulmonary malignancies.

In the images of primary lung cancer (specimens #1 and #2, Figure 3 and Figure 4), the spiculate margin between the malignant lesion and the normal area was identified as one of the specific features for distinguishing malignant pulmonary lesions in conventional CT scans. This can be observed more clearly in the volume-rendering videos (Appendix A).

Fog-like haziness, called GGO in conventional CT, was identified in the refraction-contrast SR CT images of specimen #2. This area was demarcated from the normal lung tissue; however, it contained identifiable alveolar structures (Figure 4, circles with yellow dots). The GGO was thought to be due to the lepidic pattern on pathological examination. The term lepidic refers to the tumor condition of neoplastic cells proliferating along the surface of the intact alveolar walls without stromal or vascular invasion [20].

A solitary lepidic pattern without a definite invasive component (usually shown as consolidation on X-ray imaging) is defined as adenocarcinoma in situ, which is a very early-stage lung cancer. It is less invasive; therefore, the treatment strategy tends to be less extensive [5,21]. A mixture of lepidic and other invasive subtypes are often observed in lung adenocarcinoma [6,22]. The proportions of the lepidic and other invasive components can vary in different cases. For example, specimen #1 showed cancers with only invasive components.

When GGO is found on chest CT, the proportion calculation is important because this numerical value can change the precise course of treatment. More than 50% involvement of GGO implies a better prognosis [23,24,25]; therefore, less extensive surgical excision can be considered compared to cases with less GGO or only invasive components [5,26,27].

The calculation was performed using two-dimensional findings of conventional chest CT, dividing the largest diameter of the consolidative lesion by the whole tumor size, including GGO, in the cross-section where the largest whole tumor diameter was observed [28]. Because this information was obtained from planar data, it does not reflect the actual percentage of GGO inside the entire tumor. Hence, 3D volume-rendering images from synchrotron XDFI CT can provide a solution for acquiring substantial measurements.

XDFI images from specimen #2 showed qualified contrast between the solid component, GGO in the tumor area, and normal lung parenchyma, indicating good image quality. However, in the case of specimen #1, the refraction-contrast SR images did not display the same level of contrast. In these images, the boundary of the solid part was well demarcated; however, the normal parenchyma seemed to be crowded compared with that of specimen #1. We suspected that inflation had not been properly performed before fixation. Because the lungs collapse before resection during the surgical procedure, airway instillation is essential to comprehend the intact status of the pulmonary lesions. Our institute follows the inflation fixation technique described by Hausmann [29], which is effective and simple. However, errors can occur during the detailed processing of individual samples.

An irregular spiculate margin around the tumor border, eccentric calcification, air bronchogram or bubble-like lucencies inside the tumor, and pleural involvement of pulmonary nodules are typical findings suggesting primary malignancy on conventional chest CT [30,31]. The SR images of the previous two specimens showed all these features and enabled volumetric assessment.

Specimens #3 and #4 displayed other types of lung malignancies, specifically secondary or metastatic lung cancer. We chose adenocarcinoma of colorectal origin because it is one of the most common secondary adenocarcinomas of the lung, and a differential diagnosis is often important for clinical decision making. Characteristic pathological findings of lung adenocarcinoma of colorectal origin include relatively smooth, lobulated margins, without air bronchograms or bubble-like lucencies inside the tumor area, and occasional cavitation [31,32].

Reconstructed images from specimen #3 revealed a well-demarcated, round, non-spiculate margin between the malignant and normal lesions (Figure 5b–d). The overall 3D morphology and depth in the lateral view were observed more distinctively in the volume-rendered images (Appendix A). The bubble-like lucencies identified in specimen #2 were not observed in these images. Sporadic calcification was expressed as multiple small whitish irregular dots, which were thought to be traces of secretions from glandular cells in metastatic tumors and were observed as pinkish lesions on pathological examination (Figure 5a).

Images from specimen #4 contained large pore lesions with a small proportion of malignancies. We suspected that this cavitation was filled with secretions, probably produced by malignant glandular cells in the tumor, and washed away during the preparation process. Approximately one-fifth of metastatic lung cancers of colorectal origin show cavitary lesions on both imaging and pathological examination.

Refraction-contrast SR has proven to be an effective method for soft-tissue tomographic imaging. Our study demonstrated the possibility of differential diagnosis of pulmonary malignancies using this imaging modality. It can successfully yield imaging characteristics of each pulmonary malignancy with a higher resolution. The reconstructed images enable the estimation of volumetric information, including the stereotactic GGO proportion. Clinically significant lesions can be highlighted, magnified, and reconstructed into 3D images and are therefore helpful for treatment decision making.

For the diagnosis of early-stage lung cancer, the identification of GGO lesions using X-ray imaging is invaluable. Malignant lesions containing GGO are frequently expected to exhibit lepidic patterns and cancerous conditions but are non-invasive in character [33,34]. The presence of such patterns indicates a less invasive behavior, making resection options (wedge resection or segmentectomy) feasible [34,35]. A higher proportion of GGO is generally associated with better outcomes. The current calculation method for GGO proportion comprises only two-dimensional information in one section, which could lead to misleading clinical decisions.

To assess the possibility of GGO detection using refraction-contrast SR CT, we performed 3D reconstruction imaging using Amira-Avizo software (version 2020.3, Visualization Sciences Group, Burlington, MA, USA) from the images of specimen #2. Images from the this software showed a more clearly demarcated border between the malignant and normal lung parenchyma, as well as between the GGO and the consolidative lesion inside the tumor (Appendix A).

When a pulmonary nodule found on chest CT is small (usually less than 3 cm) and the patient has a history of extra-thoracic cancer, discriminating between primary and secondary malignancies is important for further clinical decisions [36,37,38]. Surgical excision is considered when the nodule is highly suspicious for secondary lung cancer in these patients, and the diagnosis is made after surgical resection [39,40].

Histological examination was performed using a frozen biopsy analysis of a completely prepared specimen during or after surgery. Performing the analysis after surgery would ensure accurate conclusions; however, patients would have to undergo another surgery if the nodule is determined to be a primary malignancy [41]. Frozen analysis can alleviate the patient’s burden by reducing the chances of reoperation; however, there is a possibility of inconsistency between frozen and permanent diagnoses. The reported concordance between frozen permanent results ranges from 70 to 95% [42,43]. Further refinement and application of refraction-contrast SR images based on XDFI optics to 3D virtual histology could provide solutions to these problems and assist in histological decision making in the future.

Our study delineated several salient limitations. Primarily, the substantial dimensions of current synchrotron imaging apparatus render its integration within hospital infrastructure impractical. A conceivable solution could involve the juxtaposition of a medical facility with a synchrotron installation. Nonetheless, this arrangement presents logistical complexities in the transfer of specimens. The duration of transit between these sites could significantly exceed the permissible time window for frozen section analysis.

Furthermore, the imaging protocol is characterized by notable latency during the phases of acquisition, reconstruction, and analytical processing. The tasks of three-dimensional volume rendering and segmentation are discrete and occasionally prolonged activities, at times extending over multiple days. Such delays pose considerable impediments to the clinical utility of this method. As it stands, our technique requires approximately three hours to process each specimen. For clinical relevancy, this timeframe necessitates reduction to surpass that of traditional frozen section analysis.

A further constraint is the dimension of the specimen. Typically, our samples are confined to dimensions of 2 cm in width, 4 cm in height, and a depth of 3 mm. Theoretically, our methodology is capable of accommodating specimens measuring up to 4 cm in width and 6 cm in height. While this may suffice for imaging smaller lung cancers, which were less than 2 cm in size and generally indicative of early-stage disease, the development of an extended field of view is imperative for larger samples.

However, this study had several limitations. First, current equipment required for synchrotron imaging is too large to be installed in hospital buildings. Constructing a hospital next to a synchrotron facility, or vice versa, would be a possible alternative; however, there would be problems related to specimen delivery. Transportation between the two facilities would be significantly time-consuming, probably exceeding the interval necessary for frozen analysis. Considerable costs for development and adaptation are required for the clinical use of this equipment, and these expenses should be taken into account before its practical implementation.

Moreover, there are pauses in imaging acquisition, reconstruction, and analysis. The processes of 3D volume rendering and segmentation are separate events. Occasionally, these processes take several days to complete, complicating their clinical application. Our method requires approximately 3 h to image each specimen. This should be reduced to less than the time interval required for frozen section analysis to have a comparable clinical value.

Obtaining an acceptable specimen size is a problem that must be addressed. The average size of our samples was 2 cm horizontally and 4 cm vertically, with a depth of 3 mm. Theoretically, our method can acquire images from a specimen approximately 4 cm horizontally and 6 cm vertically. For small-sized lung cancers less than 2 cm (clinically early-stage lung cancer), it may be sufficient to obtain images from the resected lung tissue; however, a more extensive field of view should be developed.

Additionally, the scope of our study was somewhat limited by the narrow range of experimental specimens, which potentially impacts its clinical relevance. Our research focused on only four types of lung adenocarcinomas, comprising two primary and two secondary variants. While the findings from this experiment offer valuable insights that could inform the trajectory of future research, drawing definitive conclusions about their broader clinical applicability is currently challenging. Furthermore, the potential for subjective bias in interpreting the imaging data from this study warrants consideration. To mitigate these limitations, enhancing the diversity of our sample pool and integrating an automated imaging analysis system, particularly one utilizing deep learning algorithms based on artificial intelligence, could prove beneficial.

Recent studies on ESFR have shown remarkably higher-resolution hierarchical tomographic imaging of donated human organs [44]. The whole aspect of each human organ can be explored from the macro-to the subcellular level in one stage of imaging acquisition. Although our XDFI CT modality does not cover the entire human lung, we believe it possesses unique advantages. These include the proven comparability of the synchrotron imaging method with pathologic examination and the potential for 3D virtual histology [11,13,17,45].

Consequently, the clinical application of synchrotron radiation (SR) tomographic imaging techniques continues to face substantial challenges. Overcoming these constraints expeditiously remains a daunting endeavor. This research contributes to the ongoing efforts to ascertain the diagnostic efficacy of SR imaging in contrast to traditional X-ray imaging methods, delineating avenues for future scholarly inquiry. Notably, esteemed research institutions, including the European Synchrotron Radiation Facility (ESRF), are currently pioneering studies to adapt these technologies for clinical purposes. Consequently, it is anticipated that the near future will witness the emergence of more sophisticated and innovative approaches in this field.

## 5. Conclusions

Through dedicated and sustained research over several years, we have explored the possible potential for employing the Synchrotron XDFI-based refraction-contrast tomographic technique as a diagnostic instrument. This experimental endeavor has illuminated the technique’s nuanced capacity to differentiate between healthy and pathological lung parenchymal structures. It offers a discerning approach to the diagnosis of various lung tumor types. A distinctive advantage of this method lies in its proficiency to render three-dimensional volumetric data of pathological lesions via a reconstruction process, an achievement not paralleled by conventional X-ray imaging or isolated pathological assessments.

Despite the presence of numerous challenges yet to be surmounted, the XDFI methodology exhibits possible potential for augmenting the realm of medical X-ray imaging applications. In our recent initiative, we introduced novel imaging techniques aimed at enhancing spatial resolution. This was accomplished by positioning a scintillator in immediate proximity to a Laue angle analyzer (LAA), effectively obviating the need for X-ray interference and wavefront separation distances. Pursuant to this development, we plan to conduct further experiments with lung cancer specimens utilizing this advanced method.

## 6. Patents

Specimens used in this study were harvested from the patients who underwent surgical resections for their disease. Donations were obtained with written informed content from patients before enrollment. The patient list is described in Table 1.

## Figures and Tables

**Figure 1 cancers-16-00806-f001:**
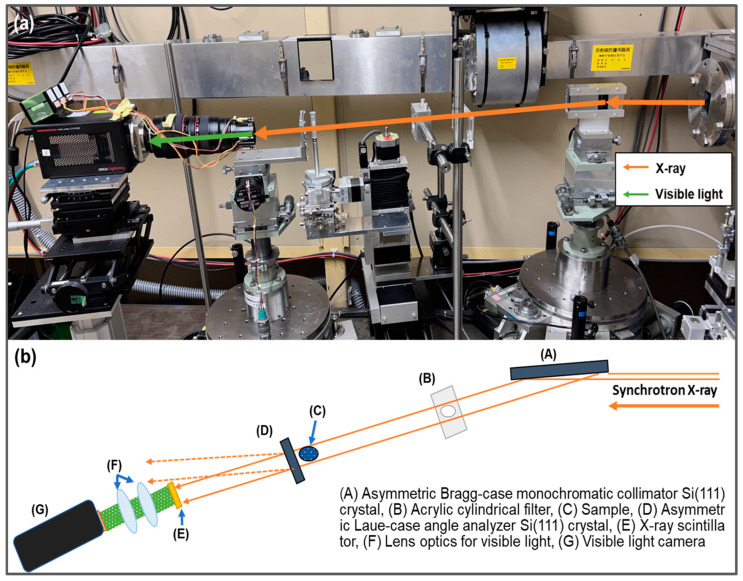
Beamline setup (**a**) and schematic view (**b**) of BL 14B in Photon Factory.

**Figure 2 cancers-16-00806-f002:**
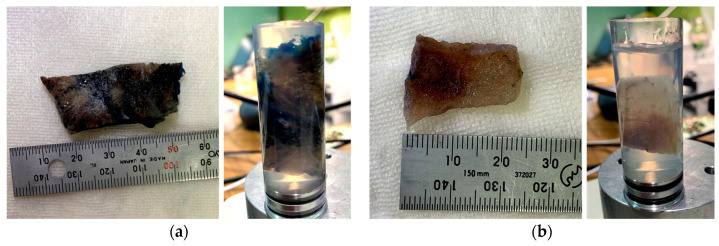
Preparation of specimens from primary lung adenocarcinoma and secondary lung adenocarcinoma (colorectal origin). (**a**) Measurement of sample #1 (**left**) and sample contained in 2 cm diameter cylinder filled with 1% agarose gel solution (**right**); (**b**) those of sample #2. Measurement (**left**) and sample contained in 2 cm diameter cylinder filled with 1% agarose gel solution (**right**); (**c**) sample #3. Measurement (**left**) and sample contained in 2 cm diameter cylinder filled with 1% agarose gel solution (**right**) and (**d**) sample #4. Measurement (**left**) and sample contained in 2 cm diameter cylinder filled with 1% agarose gel solution (**right**).

**Figure 5 cancers-16-00806-f005:**
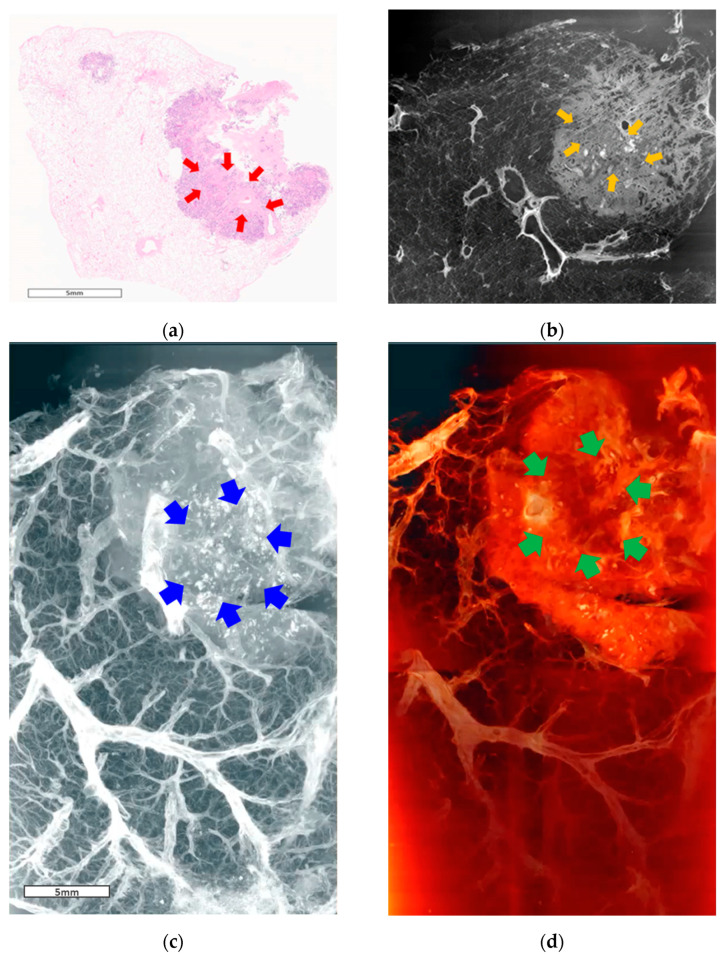
Images of specimen #3. (**a**) Pathological examination under a light microscope (scanning view) of secondary lung cancer (of colorectal origin). Smooth lobulated margins can be observed. Multiple pinkish spots (red arrows) inside the tumor area are thought to be secreted by malignant glandular cells. (**b**) One-slice image of the reconstructed tomographic volume. A similar marginal appearance was observed upon pathological examination. Dark area with irregular margin (yellow arrows) were suspected of reflecting secretions inside the tumor. (**c**) 3D image from reconstructed tomographic volume. A relatively smooth margin is observed. No tentacle-like structures are observed. Multiple whitish spots which could be calcifies lesion inside secretion were observed (blue arrows). (**d**) 3D image produced by volume rendering of the reconstructed tomographic volume. The border of the tumor is well-demarcated from the normal area. Dark reddish area (green arrows) which were suspected to be a reflection of secretion in (**a**). The whole three-dimensional aspects of the tumor lesions can be observed in the video images (Appendix A).

**Figure 6 cancers-16-00806-f006:**
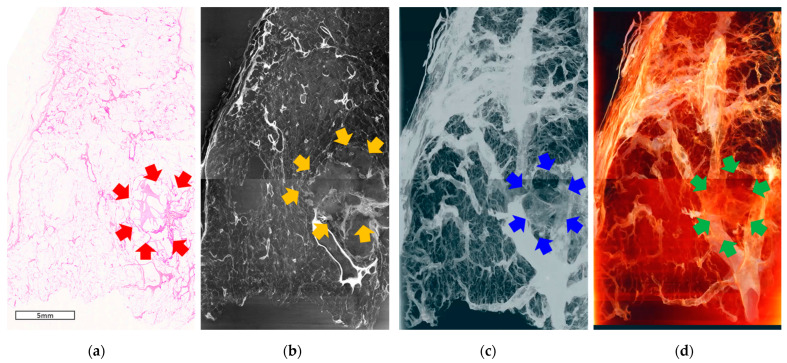
Images of specimen #4. (**a**) Pathological examination under a light microscope (scanning view) of a secondary lung cancer (of colorectal origin). A large cavitary area (red arrows) attached to a small malignant lesion can be observed. Cavitation is observed in one-fifth of metastatic cancers of colorectal origin and is thought to be caused by secretions from malignant cells. They were then washed during fixation. (**b**) One-slice image of the reconstructed tomographic volume. Cavity lesions (yellow arrows) can be observed more precisely. Contrast between the normal parenchyma was better than that in the pathological examinations. (**c**) 3D image from reconstructed tomographic volume. The demarcation of malignant lesions is relatively unclear; however, shadows of cavitary lesion were observed (blue arrows). (**d**) 3D image produced by volume rendering of the reconstructed tomographic volume. Cavitary lesion was also identified (green arrows). The three-dimensional aspects of the tumor lesions can be observed in the video images (Appendix A).

**Figure 7 cancers-16-00806-f007:**
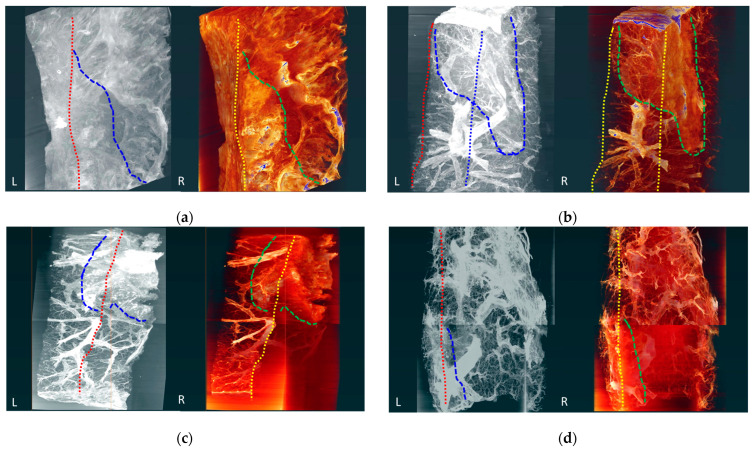
Volume rendering (L) and segmentation (R) images for each specimen. The oblique views are compared. In each image, the red dotted lines indicate the edge of specimen, and blue dotted line represents the depth of the cancerous area (L). The yellow dotted lines indicate the edge of specimen, and the green dotted lines indicate the depth of cancerous lesions (R). (**a**) Oblique slices of specimen #1 (lung adenocarcinoma without lepidic pattern). (**b**) Oblique slices of specimen #2 (lung adenocarcinoma with lepidic pattern). (**c**) Oblique slices of specimen #3 and (**d**) #4 (secondary lung adenocarcinomas from colorectal origin).

**Table 1 cancers-16-00806-t001:** Pathologic information of harvested specimens.

Number	Age	Sex	Diagnosis	Pathologic Information
#1	64	M	primary lung cancer (RLL ^1^)	adenocarcinoma, acinar-predominant, pT2bN0M0
#2	55	F	primary lung cancer (LLL ^2^)	adenocarcinoma, acinar-predominant pT1cN0M0
#3	49	F	secondary lung cancer (RLL)	adenocarcinoma (sigmoid colon origin)
#4	66	M	secondary lung cancer (RLL)	adenocarcinoma (sigmoid colon origin)

^1^ RLL; right lower lobe, ^2^ LLL; left lower lobe.

**Table 2 cancers-16-00806-t002:** Experimental conditions of BL 14B at Photon Factory.

Variables			Condition
	**incident X-ray beam**		
		X-ray energy	Monochromatic 19.8 keV
		Diffraction plane of double-crystal monochromator	Symmetric Bragg-case Si (111)
		Beam size after diffraction by MC	23^H^ × 21^V^ mm^2^
		Number of photons	Approximately 10^8^ photons/mm^2^/s
		Measurement time per sample	3 h
	**AMC** ^1^		
		Diffraction plane	Asymmetric Bragg-case Si (111)
		Thickness	20.5 mm
		Area size	124.8^H^ × 42.8^V^ mm^2^
		Asymmetric angle	5.4 degree
	**LAA** ^2^		
		Diffraction plane	Asymmetric Laue-case Si (111)
		Thickness	166 μm
		Area size	55^H’^× 50^V^ mm^2^
		Asymmetric angle	5 degrees
	**sample rotation stage**		
		Step angle	0.144 degree
		Rotation angle	360 degrees
		Number of projections	2500
	**X-ray camera**		
		Optical camera	ORCA-Lightning digital CMOS camera
			Hamamatsu Photonics K.K.
		FOV ^3^	25.3^H^´ × 14.3^V^ mm^2^
		Pixel size	5.5 μm
		X-ray scintillator	LuAG: Ce, Thickness: 100 μm
		Lens optics	85 mm, F1.2, Canon inc. (Tokyo, Japan)

^1^ AMC; asymmetrically cut monochromator collimator, ^2^ LAA; Laue-type angle analyzer, ^3^ FOV; field of view.

**Table 3 cancers-16-00806-t003:** Comparison of imaging conditions between microscopy, μ-CT ^1^ based on absorption contrast imaging, and XDFI phase-contrast imaging.

Variables	Conventional Microscopy of Stained Tissue Section	μ-CT Based on Absorption Contrast Imaging	XDFI ^2^-Based SR ^3^ Imaging
Source	Visible light	Tube X-ray	Synchrotron X-ray
Physical quantity of pixels	Color difference by tissue staining	Attenuation coefficient	Electron density
Contrast	High enough to delineate cell nuclei	Poor soft-tissue contrast	High soft-tissue contrast
Spatial resolution	Sub-μm	Sub-μm~20 μm	≈μm
		(depends on imaging condition)	data
Field of view	Several cm × several cm	Several cm × several cm	23.5 × 14.3 mm^2^
		(depends on imaging condition)	data
3D ^4^ imaging quality	Management in connection	High quality	High quality
	Between slices		
Measurement time	Long time required due to huge	10 s~several hours	3 h
	amount of slices and staining	(depends on imaging condition)	

^1^ CT; computed tomography, ^2^ XDFI; X-ray dark-field imaging, ^3^ SR; synchrotron radiation, ^4^ D; dimensional.

## Data Availability

The authors confirm that the data supporting the findings of this study are available in the Appendix A.

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
