# Peer review of "Synchrotron Radiation Refraction-Contrast Computed Tomography Based on X-ray Dark-Field Imaging Optics of Pulmonary Malignancy: Comparison with Pathologic Examination"

_cancers, 2024, doi:10.3390/cancers16040806_

Round 1

Reviewer 1 Report

Comments and Suggestions for Authors

Dear Authors,

The main concerns are given as follows:

1)      There are grammatical errors and unclear sentences in the paper. The authors require a native speaker to proofread. The authors can use the professional version of the Grammarly system.

2)      The authors should remove the words such as Background, Methods, … in the Abstract.

3)      The main contributions should be itemized at the end of the Introduction section.

4)      The Related Works section should be added after the Introduction section. The authors should review research studies by 2023 and those assigned in the Related Works section.

5)      The proposed framework should be illustrated as a figure in the Materials and Methods section.

6)      The ablation study should be performed for more experiments.

7)      The study’s strengths, weaknesses, and limitations should be written in a separate section after the discussion section.

8)      The authors should compare their proposed method with the previous methods. They need to provide the comparison Table in the Discussion section.

9)       The conclusion section should be more explained. Also, the authors should write future works in the last section.

Comments on the Quality of English Language

There are grammatical errors and unclear sentences in the paper. The authors require a native speaker to proofread. The authors can use the professional version of the Grammarly system.

Author Response

RESPOND TO THE REVIEWERS’ COMMENTS

The authors would like to sincerely appreciate the reviewers for careful review of our manuscript and providing us with their comments and suggestion. Those valuable comments were tremendously helpful for improving the quality of our manuscript. We carefully revised our previous manuscript to adjust it according to the reviewers’ advices. The following responses have been prepared to address all of the editor’s comments in a point –by-point fashion.

Reviewer 1

The main concerns are given as follows:

Comment 1

There are grammatical errors and unclear sentences in the paper. The authors require a native speaker to proofread. The authors can use the professional version of the Grammarly system.

Answer 1

I and my coauthors are sincerely grateful for the reviewer’s comments. We do feel the necessity of correcting all the grammatical errors by professional English proofreading system. We promise we will entrust our manuscript to experienced English-speaking colleagues after revising the contents.

Comment 2

The authors should remove the words such as Background, Methods, … in the Abstract.

Answer 2

Thank you for your precious comments. We realized that we didn’t remove those words like Background and Methods. We removed ‘Background’, ‘Methods’, ‘Results’, and ‘Conclusions’ in the abstract according to the reviewer’s comments.

Comment 3

The main contributions should be itemized at the end of the Introduction section.

Answer 3

We truly appreciate for the reviewer’s advice. It would be very important to describe main contributions in the end of Introduction with itemized sentences. We revised and added the itemized sentences in the introduction section according to the reviewer’s comments. Corrections were marked with yellowish color in the text.

Correction 3

We revised and appended itemized description of our experiment in the section of Introduction as the reviewer had pointed out. Added sentences were highlighted with yellowish color in the text.

The work of our experiments was described as follows. (1) acquisition of SR im-ages using XDFI optics from human lung adenocarcinomas: ether primary or secondary. (2) Specifications of secondary adenocarcinoma: acquisition of SR tomographic images of secondary adenocarcinoma originated from rectal cancer. (3) Comparison with pathology: Conducted a comparison of these images with pathological examinations. (4) Exploration of Potential: Explored the potential of refraction-contrast SR CT images in discriminating between different types of pulmonary malignancies.

Comment 4

The Related Works section should be added after the Introduction section. The authors should review research studies by 2023 and those assigned in the Related Works section.

Answer 4

We deeply appreciate for the reviewer’s precious comments. Since this experiment had been performed in 2021, we should have reviewed recent researches published after our experiment. We do apology for our mistakes and would like to add the results of our research outcomes published in March, 2023 at the end of Introduction according to the reviewer’s advice.

Correction 4

We summarized our recent research work and described them in the end of Introduction section as the reviewer’s comment. Changes were marked with yellowish color in the text.

Recently we proposed new imaging methods for improving the spatial resolution by placing a scintillator in close contact with a Laue angle analyzer (LAA), therefore could eliminate the distance required for x-ray interference and wavefront separation. Further experiments with lung cancer specimen will be proceeded using this method [19].

[19] Sunaguchi, N., et al., Superimposed wavefront imaging of diffraction-enhanced x-rays: A method to achieve higher resolution in crystal analyzer-based x-ray phase-contrast imaging. Applied Physics Letters, 2023. 122(12).

Comment 5

The proposed framework should be illustrated as a figure in the Materials and Methods section.

Answer 5

Thank you for valuable advice and we do really apologize for our mistakes. The details of framework in experimental hutch should be described in the Materials and Methods section of the manuscript. We revised and added explanation relating with the framework in the text according to the reviewer’s comment. Corrections were highlighted with yellowish color in the text.

Correction 5

We revised the Materials and Methods section, and described the illustration of framework as the reviewer had pointed out. Changes were marked with yellowish color in the text.

The imaging system consists of an asymmetrically cut monochromator collimator (AMC) made from an asymmetric Si single crystal, which described as (A) in Figure 1, an acrylic cylindrical filter (B), a sample rotation stage (C), a Laue-type angle analyzer (LAA) made from thin Si single-crystal plates (D), X-ray scintillator (E), and Lends (F) and X-ray camera (G). Because the X-ray beam is generated by a vertical wiggler, the polarization direction of the beam is perpendicular to the ground, al-lowing the crystal optics in the imaging system to be placed horizontal to the ground The experimental setup was described in Figure 1.

Comment 6

The ablation study should be performed for more experiments.

Answer 6

Thank you very much for the reviewer’s comment. The ablation therapy through percutaneously inserted devices such as radiofrequency ablation, microwave ablation or irreversible electroporation have been proved to implicate therapeutic effect on lung cancer treatment, ether primary or secondary. We indeed agreed that there should be performed ablation study in the future. However, ablation therapy has not been approved as a standard treatment for pulmonary malignancy in our country, the planning and practice of those kinds of studies seemed to be very hard currently. We promise to do our best for conducting ablation studies in our further experiments.

Comment 7

The study’s strengths, weaknesses, and limitations should be written in a separate section after the Discussion section.

Answer 7

I and my coauthors are profoundly grateful for your comment. We checked our previous manuscript and realized that more structured summary relating with the strengths, limitations and weaknesses of our study. We revised and appended the paragraphs regarding with those matters in the end of the Discussion section according to the reviewer’s advice. Changes were highlighted with yellowish color in the text.

Correction 7

We separated and rewrote the paragraphs describing the implication, limitation and obstacles we should overcome in further study in the end of Dissection section. Corrections were highlighted with yellowish color in the text.

However, this study had several limitations. First, current equipment required for synchrotron imaging is too large to be installed in hospital buildings. Constructing a hospital next to a synchrotron facility, or vice versa, would be a possible alternative; however, there would be problems related to specimen delivery. Transportation be-tween the two facilities would be significantly time-consuming, probably exceeding the interval necessary for frozen analysis. Considerable costs for development and adaptation are required for the clinical use of this equipment, and these expenses should be taken into account before its practical implementation.

Moreover, there are pauses in imaging acquisition, reconstruction, and analysis. The processes of 3D volume rendering and segmentation are separate events. Occasionally, these processes take several days to complete, complicating their clinical application. Our method requires approximately 3 h to image each specimen. This should be reduced to less than the time interval required for frozen section analysis to have a comparable clinical value.

Obtaining an acceptable specimen size is a problem that must be addressed. The average size of our samples was 2 cm horizontally and 4 cm vertically, with a depth of 3 mm. Theoretically, our method can acquire images from a specimen approximately 4 cm horizontally and 6 cm vertically. For small-sized lung cancers less than 2 cm (clinically early-stage lung cancer), it may be sufficient to obtain images from the resected lung tissue; however, a more extensive field of view should be developed.

Additionally, the scope of our study was somewhat limited by the narrow range of experimental specimens, which potentially impacts its clinical relevance. Our re-search focused on only four types of lung adenocarcinomas, comprising two primary and two secondary variants. While the findings from this experiment offer valuable insights that could inform the trajectory of future research, drawing definitive conclusions about their broader clinical applicability is currently challenging. Further-more, the potential for subjective bias in interpreting the imaging data from this study warrants consideration. To mitigate these limitations, enhancing the diversity of our sample pool and integrating an automated imaging analysis system, particularly one utilizing deep learning algorithms based on artificial intelligence, could prove beneficial.

Consequently, the clinical application of Synchrotron Radiation (SR) tomographic imaging techniques continues to face substantial challenges. Overcoming these constraints expeditiously remains a daunting endeavor. This research contributes to the ongoing efforts to ascertain the diagnostic efficacy of SR imaging in contrast to traditional X-ray imaging methods, delineating avenues for future scholarly inquiry. Notably, esteemed research institutions, including the European Synchrotron Radiation Facility (ESRF), are currently pioneering studies to adapt these technologies for clinical purposes. Consequently, it is anticipated that the near future will witness the emergence of more sophisticated and innovative approaches in this field.

Comment 8

The authors should compare their proposed method with the previous methods. They need to provide the comparison Table in the Discussion section.

Answer 8

We are truly thankful for the reviewer’ advice and indeed agree with the opinion that there must be table comparing each imaging methods. We made separate table for comparing the imaging conditions among examinations conducted using light microscopy, micro-computed tomography (CT), and synchrotron radiation (SR) tomographic images based on XDFI optics. We thought that the advantage of SR CT based on XDFI over conventional micro-CT based absorption contrast or images under light microscopy would higher resolution (around 10 μm) and higher soft tissue contrast. Comparisons were summarized in Table 3. And added in the section of Methods and Materials, which described comparison of pathologic examination and SR CT images. We realized the importance of distinguishing between pathologic examination and synchrotron radiation tomographic imaging methods. With utmost respect for the reviewer's perspective, we thought we could relocate this table to the Discussion section, should the reviewer advise that such a modification would enhance the manuscript's clarity and coherence.

Correction 8

We revised and made additional table to describe comparison between our proposed method and other conventional imaging methods according to the reviewer’s comment. This table was inserted after description of comparison with pathologic examinations in the section of Methods and Materials. Changes were marked with yellowish color in the text.

Table 3 compares the imaging conditions among examinations conducted using light microscopy, micro-CT, and synchrotron radiation based on XDFI optics.

Variables

Conventional Microscopy

 of stained tissue section

Micro CT based on

absorption contrast imaging

XDFI2 based

SR3 imaging

   Source

Visible light

Tube X-ray

Synchrotron X-ray

Physical quantity of Pixel

Color difference by tissue staining

Attenuation Coefficient

Electron density

Contrast

High enough to delineate cell nuclei

Poor soft tissue contrast

High soft tissue contrast

Spatial resolution

Sub μm

Sub μm ~ 20 μm

μm

(depends on imaging condition)

data

Field of View

Several cm x Several cm

Several cm x Several cm

23.5 x 14.3 mm2

(depends on imaging condition)

data

3D4 imaging quality

Management in connection

High quality

High quality

Between slices

Measurement time

Long time required due to huge

10 sec ~several hours

3 hours

amount of slices and staining

(depends on imaging condition)

* 1CT; Computed Tomography, 2XDFI; X-ray Dark Field Imaging, 3SR; Synchrotron Radiation, 4D; dimensional

Comment 9

The conclusion section should be more explained. Also, the authors should write future works in the last section.

Answer 9

Thank you for your thoughtful advice. We do really agree with the reviewer’s opinion that more refinement and additional phrase relating with our further works. We revised and changed our Conclusion section according to the reviewer’s comment. We believe this endeavor could improve our manuscript. Corrections were marked with yellowish color in the text.

Correction 9

We checked and reformed the contents of our conclusion section as the reviewer had pointed out. Corrections were highlighted with yellowish color in the text.

Through dedicated and sustained research over several years, we have explored the possible potential for employing the Synchrotron XDFI-based refraction-contrast tomographic technique as a diagnostic instrument. This experimental endeavor has illuminated the technique's nuanced capacity to differentiate between healthy and pathological lung parenchymal structures. It offers a discerning approach to the diagnosis of various lung tumor types. A distinctive advantage of this method lies in its proficiency to render three-dimensional volumetric data of pathological lesions via a reconstruction process, an achievement not paralleled by conventional X-ray imaging or isolated pathological assessments.

Despite the presence of numerous challenges yet to be surmounted, the XDFI methodology exhibits possible potential for augmenting the realm of medical X-ray imaging applications. In our recent initiative, we introduced novel imaging techniques aimed at enhancing spatial resolution. This was accomplished by positioning a scintillator in immediate proximity to a Laue angle analyzer (LAA), effectively obviating the need for x-ray interference and wavefront separation distances. Pursuant to this development, we plan to conduct further experiments with lung cancer specimens utilizing this advanced method.

The authors sincerely appreciate the reviewers’ valuable comments, and we carefully revised our manuscript according to the editor’s advice. We believe we did our best to improve the quality of our manuscript, and wish our revision have better achievement. If there were anything to be corrected or appended, please let us know and we promise we will make every effort to revise again.

Reviewer 2 Report

Comments and Suggestions for Authors

1. The paper does not adequately compare the new imaging method with existing diagnostic tools. A more comprehensive comparison, including sensitivity, specificity, and diagnostic accuracy, would be essential to establish the method's relative effectiveness in lung cancer diagnosis.

2. The study seems to have a limited sample size and lacks diversity in terms of the types of lung cancer specimens examined. A broader range of samples, including various stages and types of lung cancers, would be necessary to validate the effectiveness of the imaging technique universally.

3. The paper does not address the practical aspects of implementing this technology in a clinical setting, such as the availability, cost, and operational requirements of the synchrotron radiation equipment. This information is crucial for assessing the feasibility of widespread clinical adoption. 

4. The study appears to be cross-sectional and does not provide longitudinal data to assess how well this imaging technique tracks the progression or regression of lung cancer over time. Longitudinal studies would be invaluable in understanding the full clinical utility of this method. I

5. The paper may not provide enough technical detail about the refraction-contrast computed tomography method, such as specific parameters used, which is essential for reproducibility and peer evaluation.

6. The study might not sufficiently address the potential for subjective bias in interpreting the imaging results. Incorporating automated or semi-automated image analysis methods could reduce this bias and provide more objective results.

7. While the initial results are promising, the paper lacks information on further validation studies needed before this method can be considered for clinical use. Multi-center trials and independent validations are necessary to establish reliability and generalizability.

Comments on the Quality of English Language

Extensive editing of the English language is required.

Author Response

RESPOND TO THE REVIEWERS’ COMMENTS

The authors would like to sincerely appreciate the reviewers for careful review of our manuscript and providing us with their comments and suggestion. Those valuable comments were tremendously helpful for improving the quality of our manuscript. We carefully revised our previous manuscript to adjust it according to the reviewers’ advices. The following responses have been prepared to address all of the editor’s comments in a point –by-point fashion.

Reviewer 2

Comment 1

The paper does not adequately compare the new imaging method with existing diagnostic tools. A more comprehensive comparison, including sensitivity, specificity, and diagnostic accuracy, would be essential to establish the method's relative effectiveness in lung cancer diagnosis.

Answer 1

My coauthor and I extend our sincere gratitude for the reviewer's insightful comments. We concur with your perspective regarding the necessity of a thorough comparison between Synchrotron Radiation (SR) tomographic images and conventional imaging to evaluate the relative effectiveness of SR imaging methods in diagnosing lung cancer. Indeed, this comparison represents a critical initial step for clinical application. However, our current imaging method faces significant limitations concerning sample size. The sample stage and field of view (FOV) of our optics are optimally utilized only when the sample size is 2cm or smaller. We employed specimens of this size, excised from resected lung cancers, which were remnants following all requisite diagnostic procedures overseen by a specialized pathologist. These tissues were processed conventionally, fixed in 10% neutral-buffered formalin solution, and serially sectioned to a depth of 3mm. We recognize that images from conventional CT scans may not accurately represent the structure of these processed specimens. Consequently, we propose a comparison of the histological features from pathological images with those from SR imaging.

Additionally, our imaging method is constrained by other limitations. The time required for imaging reconstruction can vary considerably, ranging from several hours to days. These limitations pose significant challenges in transitioning this method from the research bench to clinical bedside applications. We acknowledge that our initial description of these limitations was insufficient. Therefore, in response to the reviewer's comments, we intend to revise the Discussion section of our manuscript to articulate these constraints more explicitly. We also aim to clarify the limitations of our study in the context of clinical application.

Correction 1

We revised the paragraphs describing the implication, limitation and obstacles we should overcome in further study in the end of Dissection section. Corrections were highlighted with yellowish color in the text.

However, this study had several limitations. First, current equipment required for synchrotron imaging is too large to be installed in hospital buildings. Constructing a hospital next to a synchrotron facility, or vice versa, would be a possible alternative; however, there would be problems related to specimen delivery. Transportation be-tween the two facilities would be significantly time-consuming, probably exceeding the interval necessary for frozen analysis. Considerable costs for development and adaptation are required for the clinical use of this equipment, and these expenses should be taken into account before its practical implementation.

Comment 2

The study seems to have a limited sample size and lacks diversity in terms of the types of lung cancer specimens examined. A broader range of samples, including various stages and types of lung cancers, would be necessary to validate the effectiveness of the imaging technique universally.

Answer 2

We greatly appreciate the reviewer's comment. We acknowledge that the lack of diversity and the restricted number of samples represent significant limitations of our study. To discuss clinical implications more effectively, it is necessary to expand the range of specimens in our research. We plan to conduct further experiments with other types of lung cancer specimens, including squamous cell carcinoma and neuroendocrine tumors. However, the acquisition, reconstruction, and interpretation of tomographic three-dimensional images using synchrotron radiation is a time-consuming process, and this has been a major obstacle in advancing our study. We assure the reviewer of our continued efforts to enhance the diversity of our samples and to validate the feasibility of this imaging tool.

Also, we would like add what the reviewer had pointed out as one of limitations. Correctios were highlighted with yellowish color in the text.

Correction 2

We revised and added a paragraph describing the limitation regarding with lack of diversity in the types of specimens we used according to the reviewer’s comment. Corrections were marked with yellowish color in the text.

Additionally, the scope of our study was somewhat limited by the narrow range of experimental specimens, which potentially impacts its clinical relevance. Our re-search focused on only four types of lung adenocarcinomas, comprising two primary and two secondary variants. While the findings from this experiment offer valuable insights that could inform the trajectory of future research, drawing definitive conclusions about their broader clinical applicability is currently challenging. Further-more, the potential for subjective bias in interpreting the imaging data from this study warrants consideration. To mitigate these limitations, enhancing the diversity of our sample pool and integrating an automated imaging analysis system, particularly one utilizing deep learning algorithms based on artificial intelligence, could prove beneficial.

Comment 3

The paper does not address the practical aspects of implementing this technology in a clinical setting, such as the availability, cost, and operational requirements of the synchrotron radiation equipment. This information is crucial for assessing the feasibility of widespread clinical adoption.

Answer 3

Thank you very much for the reviewer’s comment. As the reviewer had pointed out, we should have described the practical aspects of clinical implementation as the reviewer had pointed out. Practically direct clinical application of synchrotron radiation tomographic imaging methods is severely restricted because of the substantial cost for developing devices, challenges in constructing in-hospital equipment, and exponential time for high-resolution three-dimensional reconstruction. Although there has been enormous endeavor to realize the adaptation to clinical fields, still there are lots of obstacles to overcome practical constraints. However, information relating with clinical application is important, as the reviewer had mentioned, we thought that we should discuss the clinical restrictions as substantial limitations in the section of Discussion. We revised and rewrote the practical difficulties relating with cost, availability of equipment, and time consumption according the reviewer’s advice. Changes were marked with yellowish color in the section of Discussion.

Comment 4

The study appears to be cross-sectional and does not provide longitudinal data to assess how well this imaging technique tracks the progression or regression of lung cancer over time. Longitudinal studies would be invaluable in understanding the full clinical utility of this method.

Answer 4

I and my coauthors are profoundly grateful for your advice. We are thoroughly consent to the reviewer's astute observation regarding the imperative of longitudinal data to comprehensively ascertain the clinical efficacy of our imaging technique. Regrettably, the scope of our research, confined to the analysis of tomographic images from surgically excised human lung specimens, precludes the acquisition of such longitudinal data. This limitation stems from our inability to monitor the temporal evolution of lung lesions, including their progression or regression. Furthermore, the current state of synchrotron radiation tomographic apparatus renders it unsuitable for application in living human subjects. Consequently, it is beyond our capacity to furnish the requisite longitudinal synchrotron radiation tomographic imagery of consistent pulmonary malignancies. Nonetheless, we remain hopeful about the prospective advancements in this technology, anticipating its eventual modification for applicability in living human subjects

Comment 5

The paper may not provide enough technical detail about the refraction-contrast computed tomography method, such as specific parameters used, which is essential for reproducibility and peer evaluation

Answer 5

We are truly thankful for the reviewer’ comment and would like to sincerely apologize for the lack of technical detail about the refraction-contrast computed tomography method, which is essential for which is essential for reproducibility and peer evaluation. We revised the paragraphs relating with Figure 1, Contents of Table 2 and made separate table (Table 3) for comparing the imaging conditions among examinations conducted using light microscopy, micro-computed tomography (CT), and synchrotron radiation (SR) tomographic images based on XDFI optics according to the reviewer had pointed out. We believed those revision would compensate the lack of enough information relating with our technical details.

Correction 5

We checked and compensated the description relating with illustrating framework of Figure 1, revised and corrected the errors in Table 2, and added Table 3 in the Materials and Methods section as the reviewer had pointed out. Changes were marked with yellowish color in the text.

  • In the subtitle 2. X-ray source and experimental setup of the Materials and Methods section, we revised the sentence relating with illustrating framework of Figure 1.

The imaging system consists of an asymmetrically cut monochromator collimator (AMC) made from an asymmetric Si single crystal, which described as (A) in Figure 1, an acrylic cylindrical filter (B), a sample rotation stage (C), a Laue-type angle analyzer (LAA) made from thin Si single-crystal plates (D), X-ray scintillator (E), and Lends (F) and X-ray camera (G). Because the X-ray beam is generated by a vertical wiggler, the polarization direction of the beam is perpendicular to the ground, allowing the crystal optics in the imaging system to be placed horizontal to the ground The experimental setup was described in Figure 1.

  • In the subtitle 3. Acquisition and reconstruction of images of the Materials and Methods section, we revised and corrected the errors in Table 2.

Table 2. Experimental condition of BL 14B at Photon Factory

Variables

Condition

Incident X-ray beam

X-ray energy

Monochromatic 19.8 keV

Diffraction plane of double crystal monochromator

Symmetric Bragg-case Si (111)

Beam size after diffracted by MC

23H x 21V mm2

Number of photons

Approximately 108 photons/mm2/sec

Measurement time per one sample

3 hours

AMC1

Diffraction plane

Asymmetric Bragg-case Si (111)

Thickness

20.5 mm

Area size

124.8H x 42.8V mm2

Asymmetric angle

5.4 degree

LAA2

Diffraction plane

Asymmetric Laue-case Si (111)

Thickness

166μm

Area size

55H ´x 50V mm2

Asymmetric angle

5 degrees

Sample rotation stage

Step angle

0.144 degree

Rotation angle

360 degrees

Number of projections

2500

X-ray Camera

Optical camera

ORCA-Lightning digital CMOS camera

Hamamatsu Photonics K.K.

FOV3

25.3H ´x14.3V mm2

Pixel size

5.5 μm

X-ray scintillator

LuAG: Ce, Thickness: 100 μm

Lens optics

85 mm, F1.2, Canon inc.

1 AMC; Asymmetrically cut Monochromator Collimator, 2 LAA; Laue-type angle analyzer, 3 FOV; field of view

  • In the subtitle of 4. Comparison with pathologic examinations of the Materials and Methods section, we added a new table for comparison of each imaging methods.

Table 3 compares the imaging conditions among examinations conducted using light microscopy, micro-CT, and synchrotron radiation based on XDFI optics.

Variables

Conventional Microscopy

 of stained tissue section

Micro CT based on

absorption contrast imaging

XDFI2 based

SR3 imaging

   Source

Visible light

Tube X-ray

Synchrotron X-ray

Physical quantity of Pixel

Color difference by tissue staining

Attenuation Coefficient

Electron density

Contrast

High enough to delineate cell nuclei

Poor soft tissue contrast

High soft tissue contrast

Spatial resolution

Sub μm

Sub μm ~ 20 μm

μm

(depends on imaging condition)

data

Field of View

Several cm x Several cm

Several cm x Several cm

23.5 x 14.3 mm2

(depends on imaging condition)

data

3D4 imaging quality

Management in connection

High quality

High quality

Between slices

Measurement time

Long time required due to huge

10 sec ~several hours

3 hours

amount of slices and staining

(depends on imaging condition)

* 1CT; Computed Tomography, 2XDFI; X-ray Dark Field Imaging, 3SR; Synchrotron Radiation, 4D; dimensional

Comment 6

The study might not sufficiently address the potential for subjective bias in interpreting the imaging results. Incorporating automated or semi-automated image analysis methods could reduce this bias and provide more objective results.

Answer 6

Thank you very much for the precious comment. As the reviewer had pointed out, this study did not tell the possibility of subjective bias in interpreting our results. Computerized analysis system such as deep learning algorithm based on artificial intelligence should be considered to be adopted in future studies. For this, we thought that expanding the diversity of imaging samples is the first goal to be solved, as the reviewer had proposed in previous comment. While the outcomes of this experiment may guide the direction toward our future research should go, drawing confident conclusions about their applicability in clinical settings remains challenging. And the possibility of subjective bias in the interpretation of imaging data from this study. should be considered and described as limitations in the manuscript. Increasing diversity of samples and adoption of automated imaging analysis system deep learning algorithm based on artificial intelligence could be solution as the reviewer had mentioned. We would like to describe potential of subjective bias as one of the limitations in the section of Discussion. Corrected and added sentences were marked with yellowish color in the text.

Correction 6

We revised and appended sentences addressing the possible subjective bias in the interpretation of our imaging data as limitations in the section of Discussion. Changes were highlighted in yellowish color in the text.

Additionally, the scope of our study was somewhat limited by the narrow range of experimental specimens, which potentially impacts its clinical relevance. Our re-search focused on only four types of lung adenocarcinomas, comprising two primary and two secondary variants. While the findings from this experiment offer valuable insights that could inform the trajectory of future research, drawing definitive conclusions about their broader clinical applicability is currently challenging. Further-more, the potential for subjective bias in interpreting the imaging data from this study warrants consideration. To mitigate these limitations, enhancing the diversity of our sample pool and integrating an automated imaging analysis system, particularly one utilizing deep learning algorithms based on artificial intelligence, could prove beneficial.

Comment 7

While the initial results are promising, the paper lacks information on further validation studies needed before this method can be considered for clinical use. Multi-center trials and independent validations are necessary to establish reliability and generalizability.

Answer 7

My coauthors and I are profoundly grateful for the reviewer's insightful comments. We unanimously concur that our study currently lacks sufficient data to consider clinical applications of our imaging methods and recognize the necessity of conducting additional experimental studies for proper validation. We respectfully request the reviewer's understanding regarding the limitations of our study, which arise from the analysis being confined to only four types of lung adenocarcinomas conducted in an experimental hutch located within a synchrotron facility, distant from any hospital setting.

For several decades, we have endeavored to extract meaningful clinical information using synchrotron radiation devices equipped with X-ray Dark Field Imaging (XDFI) optics. We fully acknowledge the need for extensive further research to achieve tangible clinical relevance. At present, we are committed to enhancing the diversity of our specimen pool and incorporating a computerized imaging analysis system, as emphatically suggested by the reviewer. Additionally, we recognize the importance of collaborative efforts with various institutes in this domain to ensure independent validation of our data, thereby establishing its reliability and generalizability. We would like to describe the lack of information on further validation studies needed before this method can be considered for clinical use in our study as one of limitations in the ends of Discussion section. Corrections were marked with yellowish color in the text.

Corrections 7

We revised and rewrote sentences describing the lack of information for proper validation in our study, and necessity for further study as limitations in the section of Discussion according to the reviewer’s comment. Corrections were highlighted with yellowish color in the text.

Consequently, the clinical application of Synchrotron Radiation (SR) tomographic imaging techniques continues to face substantial challenges. Overcoming these constraints expeditiously remains a daunting endeavor. This research contributes to the ongoing efforts to ascertain the diagnostic efficacy of SR imaging in contrast to traditional X-ray imaging methods, delineating avenues for future scholarly inquiry. Notably, esteemed research institutions, including the European Synchrotron Radiation Facility (ESRF), are currently pioneering studies to adapt these technologies for clinical purposes. Consequently, it is anticipated that the near future will witness the emergence of more sophisticated and innovative approaches in this field.

The authors sincerely appreciate the reviewers’ valuable comments, and we carefully revised our manuscript according to the editor’s advice. We believe we did our best to improve the quality of our manuscript, and wish our revision have better achievement. If there were anything to be corrected or appended, please let us know and we promise we will make every effort to revise again.

Round 2

Reviewer 2 Report

Comments and Suggestions for Authors

The authors have addressed all of the comments successfully. I have no further comments on this paper.

Comments on the Quality of English Language

Extensive editing of the English language is required.